# Study on Pore Structure Evolution Characteristics of Weakly Cemented Sandstone under Freeze–Thaw Based on NMR

**Jian Lin** [1,2], **Yi Yang** [1,2,*], **Jianchao Yin** [1,2], **Yang Liu** [1,2] and **Xiangwei Li** [1,2]

1   Anhui Provincial Key Laboratory of Building Structure and Underground Engineering,
    Anhui Jianzhu University, Hefei 230601, China
2   College of Civil Engineering, Anhui Jianzhu University, Hefei 230601, China
*   Correspondence: yangyi@stu.ahjzu.edu.cn

**Abstract:** Taking saturated, weakly cemented sandstone as the research object, nuclear magnetic resonance (NMR) tests were performed before and after six freeze–thaw cycles without water replenishment in order to study and reveal the evolution characteristics of the pore structure of weakly cemented sandstone under a freeze–thaw cycle. The evolution of pore structure under repeated freeze–thaw cycles was studied using $T_2$ fractal theory and spectral peak analysis. The results show that the evolution of the pore structure of weakly cemented sandstone can be divided into three stages during the freeze–thaw cycle. In stage 1, the rock skeleton can still significantly restrict frost heave, and the effect of rock pore expansion occurs only on the primary pore scale, primarily in the transformation between adjacent scales. In stage 2, as the restraint effect of the skeleton on frost heave decreases, small-scale secondary pores are gradually produced, pore expansion occurs step by step, and its connectivity is gradually enhanced. In stage 3, as rock pore connectivity improves, the effect of pore internal pressure growth in the freezing process caused by water migration is weakened, making it impossible to break through the skeleton constraint. Thus, it becomes difficult for freezing and thawing to have an obvious expansion effect on the rock pore structure. The strength of the freeze–thaw cycle degradation effect is determined by the effect of the rock skeleton strength under the freeze–thaw cycles and the connectivity of small-scale pores in the rock. The lower the strength of the rock skeleton, the worse the connectivity of pores, and the more obvious the freeze–thaw degradation effect, and vice versa.

**Keywords:** weakly cemented sandstone; NMR; freeze thaw cycle; fractal characteristics; $T_2$ sub peak

## 1. Introduction

The mechanism of rock freeze–thaw deterioration has long been a focus of study in the field of frozen soil engineering. The deterioration effect of freezing and thawing is more visible in weakly cemented sandstone, a rock with high frost sensitivity. The internal cause of freeze–thaw damage to rock is the evolution of the pore structure of rock following the freeze–thaw cycles. As a result, it is critical to investigate the pore evolution characteristics of weakly cemented sandstone during multiple freeze–thaw cycles for understanding the mechanism of freeze–thaw deterioration.

Many domestic and foreign researchers have conducted extensive studies on the micro-pore structure of rocks using NMR (nuclear magnetic resonance), CT (computed tomography), MIP (mercury intrusion porosimetry), and other technologies. For example, NMR technology is used to obtain the pore structure characteristics of tight sandstone, the fractal theory is used to reveal the irregularity and heterogeneity of pore roaring structures, and the differences between the characteristics of different types of pore structures are explored [1–5]. Additionally, the anisotropy of the pore structure of a certain hydrostatic pressure is expressed by the anisotropy of elastic wave velocity [6]. On the basis of CT scanning, the distribution characteristics of rock pore grids are quantitatively analyzed

by introducing complex grid theory, and the connectivity changes of rock pore grids are studied [7]. Based on MIP tests and image analysis technology, the fractal characteristics of tight sandstone's pore structure are studied [8]. Yang [9] used NMR technology to study the fractal characteristics and internal pore distribution characteristics of rocks under different excitation conditions. Based on fractal dimension theory, Ding [10] constructed the correlation between particle pore distribution characteristics and mechanical strength. Jia [11] conducted static and dynamic compression tests on freeze–thaw damaged sandstone, and studied the effects of freeze–thaw cycles and strain rates on the mechanical properties and fractal dimension characteristics of sandstone. Huang [12] studied the pore structure of freeze–thaw sandstone by using fractal dimension, and established the relationship between the change in fractal dimension and strength. Niu [13] conducted a freeze–thaw cycle test on sandstone using NMR technology, and studied the change in the microstructure of the sandstone specimen caused by freeze–thaw cycles. Wang [14] conducted a freeze–thaw cycle test on rock and studied the change in rock pore structure.

The quantitative characterization of rock pore structure has become a hot and difficult topic in the field of rock meso-damage research, and many researchers have conducted exploratory studies. Liu [15] used CT scanning technology to study the failure mechanism of sandstone under freezing and thawing, and discovered that it was caused by failure of the initial structure or a defect of the internal structure. Zhou [16] used fractal theory to study the relative difference in the pore structure of Triassic glutenite, and proposed a new idea about quantitative characterization of heterogeneous pore structure. Zhao [17] adopted an atomic force microscope and image processing to quantitatively characterize the pore structure of coal and shale reservoirs. Furthermore, Zhang [18] employed fractal theory to quantitatively characterize the pore structure of different types of shale. Liu [19] quantitatively characterized the dynamic fragmentation of freeze–thaw sandstone with the help of fractal theory, and established the relationship between fractal dimension, freeze–thaw cycles, and impact strength. Abdolghanizadeh [20] studied the influence of freeze–thaw cycles and freezing temperature on the fracture toughness of sandstone, and determined the damage caused by freeze–thaw cycles with CT scanning. Sun [21,22] used image analysis technology to quantitatively characterize the microscopic pore structure characteristics of shale.

As a result, relevant technology has gradually become an important means of conducting freeze–thaw damage research. For example, Jia [23] and Li [24] conducted freeze–thaw cycling tests on tight sandstone using NMR, and concluded that the change in the pore structure of a rock is primarily caused by the expansion of nanopores, micropores, and the generation of secondary nanopores. Liu [25] adopted NMR to investigate the influence of different initial damage degrees on various indicators of rock mass, and Hou [26] discovered that as the number of freeze–thaw cycles increases, so does the surface roughness of pores and pore roaring structures. In addition, Yuan [27] performed NMR tests on anthracite and found that the rate of fracture development is inversely proportional to the number of liquid nitrogen freeze–thaw cycles. Furthermore, early research in the field of rock freeze–thaw damage showed that the elastic wave velocity of the rock decreased as the number of freeze–thaw cycles increase [28]. Li [24] carried out NMR tests on sandstone based on the diffusion double electric layer theory, and obtained the meso-structural characteristics of sandstone under freeze–thaw action. Based on the test results, Jia [29–32] established the damage evolution equation of sandstone under the action of freeze–thaw cycling, and analyzed the possible dominant action mechanism of sandstone's internal pores.

In conclusion, the above studies demonstrated that rock freeze–thaw damage is related to pore structure evolution, but the establishment and verification of freeze–thaw damage models still lack a solid experimental basis. To accurately reveal the freeze–thaw deterioration mechanism, quantitative characterization of pore structure development characteristics under freeze–thaw cycles is required. As a result, this paper uses weakly cemented sandstone as the research object, and performs an NMR test on it during repeated freezing and thawing cycles. It investigates the evolution law of its pore structure under

repeated freezing and thawing on the basis of $T_2$ fractal theory, combined with the peak analysis method of the spectrogram, in order to provide a solid test basis for establishing its freezing and thawing damage model.

## 2. Test Materials and Methods

The rock sample for this test was obtained from the Lijiagou Coal Mine in Yulin City, Shaanxi Province. It is, specifically, weakly cemented sandstone of the Jurassic Anding Formation, and its saturated color is brown. As shown in Figure 1, the rock samples were processed by in situ drilling into cylindrical samples with a diameter of 25 mm and a height of 50 mm. The sampled wave velocity, dry density, and saturated volume moisture content were in the ranges of 676~806 m/s, 2.09~2.18 g/cm$^3$, and 12.97%~13.41%, respectively. Table 1 shows the basic physical parameters of the selected test objects (Table 1).

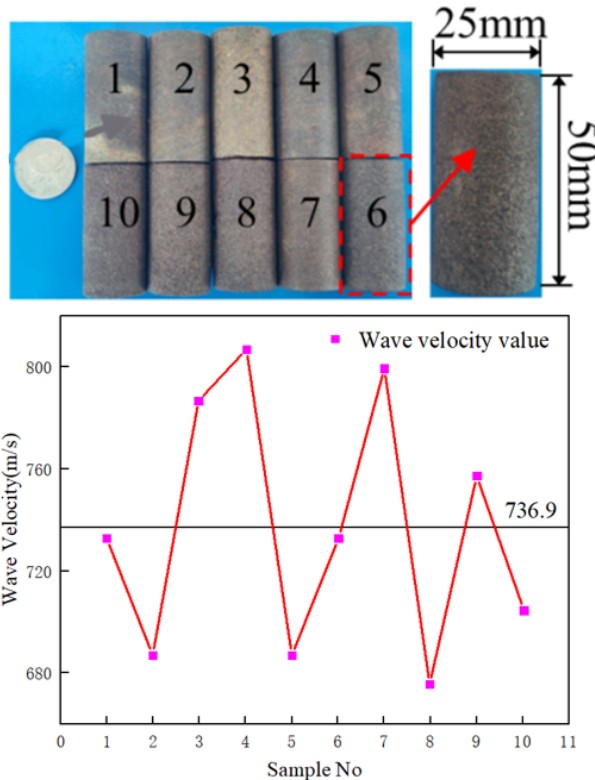

**Figure 1.** Test object and foundation parameters.

**Table 1.** Basic physical properties of rock samples.

| Wave Velocity (m/s) | Saturation Density (g/cm$^3$) | Dry Density (g/cm$^3$) | Saturated Volume Water Content (%) |
|---|---|---|---|
| 736.9 | 2.23 | 2.13 | 13.35 |

This drying test was conducted in batches, with each drying time being short. The $T_2$ spectrum was collected after each drying by combining it with NMR, and $T_2$ cut-off values were calculated. To avoid the loss of bound water caused by too high a test temperature and too long a drying time, the sample was dried for 10 min at 85 °C. Figure 2 depicts the test path and the test results. Figure 2 shows that the moisture content and drying times changed exponentially, with the moisture content nearly reaching a minimum in the last two drying tests.

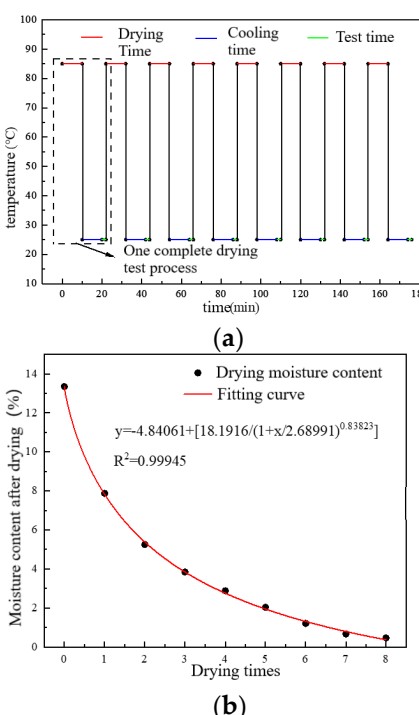

(a)

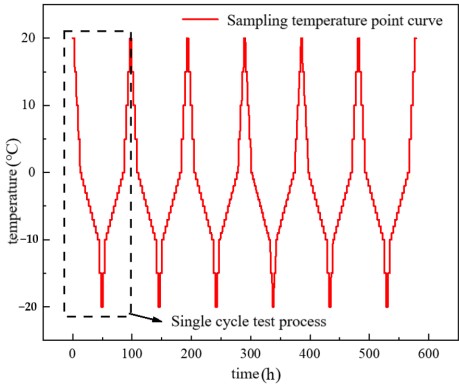

(b)

**Figure 2.** (**a**) Drying test path; (**b**) moisture content of successive drying results.

To study the damage mechanism of sandstone micropores under multiple freeze–thaw cycles with temperatures ranging from −20 °C to 20 °C, six consecutive freeze–thaw cycles were conducted. Figure 3 depicts the temperature gradient in one single cycle.

**Figure 3.** Temperature gradient in the test.

The main process of the test is as follows: (1) vacuum saturation: the sample was placed in the vacuum pressurizing saturation device, vacuumed with a pressure value of −0.1 MPa, and then saturated with a pressure of 0.5 MPa for 24 h. (2) NMR equipment calibration: a low temperature probe was chosen, then a standard sample with a different water content was used for parameter debugging and calibration to determine the relationship between NMR signal and water content. (3) NMR test with freeze–thaw cycling: the tested sample was placed in the cryogenic probe and sealed, then the cryogenic control system was connected for six repeated tests.

## 3. Analysis of Pore Structure of Rock under Freeze–Thaw Cycles

### 3.1. Variation Rule of Pores at Different Scales

The original pore types were divided into 0~3 ms, 3~33 ms, and 33~505 ms [33,34], based on the measured $T_2$ spectrum of the initial state and the $T_2$ spectrum after each cycle,

as shown in Figure 4. The proportions of micropores, fine pores, macropores, and free water before and after the freezing and thawing cycles are shown in Figure 5.

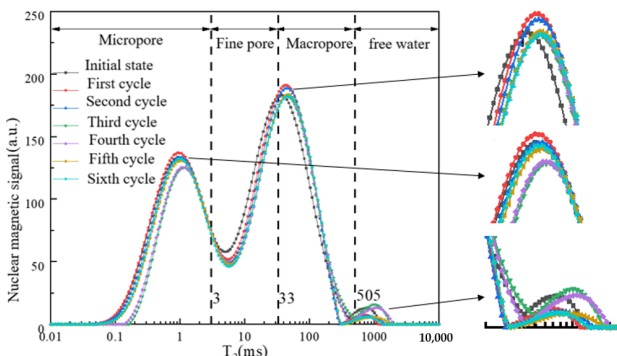

**Figure 4.** T$_2$ distribution before and after freeze–thaw cycles.

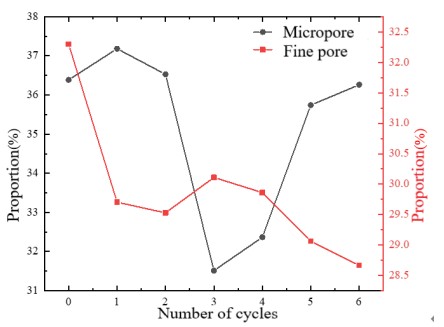
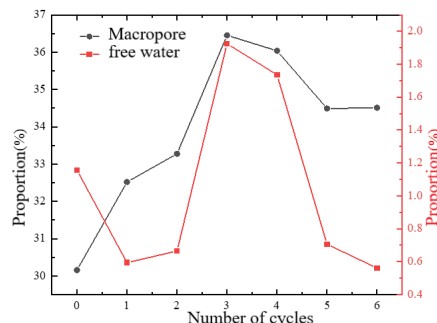

**Figure 5.** Evolution of the proportion of micro-, fine, and large pores, as well as free water, before and after freeze–thaw cycles.

According to the above experimental results, the initial proportions of micropores, fine pores, and macropores were greater than 30%, while the initial proportion of free water was less than 1.2%, and the proportions of the four types of pores changed dramatically during the third freezing–thawing cycle.

The test results are classified into three stages:

(1) After the first two freeze–thaw cycles, the macropore clearly increases, while the fine pore obviously decreases, but the micropore changes slightly; this demonstrates that, due to the constraints of the rock skeleton, there was no strong frost heave effect on micropores after the first two freeze–thaw cycles of sandstone, but there was a large degree of expansion of fine pores to large pores.

(2) After the third freezing–thawing, the macropores increased significantly while the fine pores increased slightly, although the changes followed the same trend and the micropores decreased significantly. It can be seen that frost heave broke through the constraints of the rock skeleton and expanded from micropores to fine pores and macropores in the second to third cycle stage.

(3) In the subsequent freeze–thaw cycling, the large and small pores decreased in the same way, while the micropores increased significantly; this could be due to the fact that the constraints of the rock skeleton are broken, resulting in a large number of secondary micropores and reducing the proportion of fine and large pores.

### 3.2. Change Rule of Bound and Free Pores

The T$_2$ cut-off value was calculated using the T$_2$ distribution curve of NMR at the time of complete saturation of the sandstone and the first drying. To obtain the accumulation curve, the nuclear magnetic signals in the two states were accumulated, and a horizontal line was drawn from the maximum value of the first drying accumulation so that this

horizontal line intersected with the accumulation curve in a fully saturated state. The abscissa value of the intersection point is the $T_2$ cut-off value, which was calculated to be 5.264 ms, as shown in Figure 6 below.

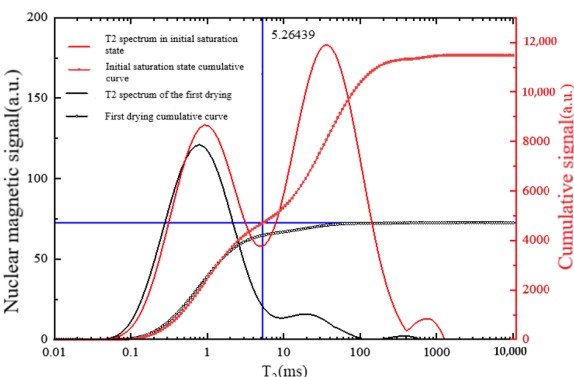

**Figure 6.** $T_2$ cutoff calculated by the drying method.

Figure 6 depicts the proportion change of the bound hole and the free hole during freeze–thaw cycling. The third freeze–thaw cycle was used as the dividing point between the two change trends, resulting in two stages. In each of the six freeze–thaw cycles, the proportion of seepage holes was always greater than 58%.

It can be seen from Figure 7 that the test results can be divided into three stages:

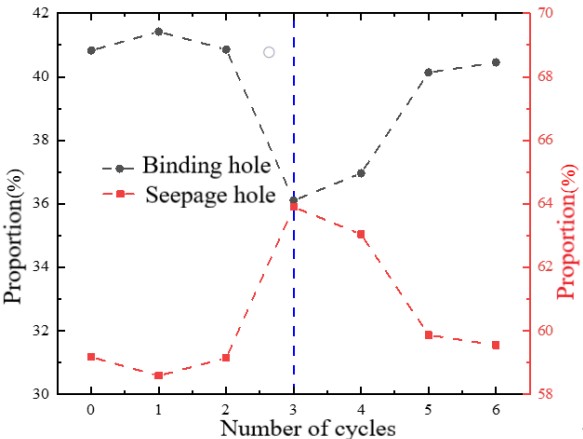

**Figure 7.** Evolution of bound pore and free pore proportions.

(1) The proportions of the bound hole and the seepage hole fluctuated in a small range after the first two freeze–thaw cycles, with a fluctuation range of 0.5%. This demonstrated that there was no obvious change in the structure of the two at this point, or that the change occurred only internally.

(2) The proportion of seepage holes increased significantly after the third cycle, while the proportion of bound holes decreased significantly. This means that the constraint of the particle skeleton of the rock was broken in the second to third cycle, and the bound hole expanded to the seepage hole.

(3) In the following cycles, the proportion of seepage holes decreased while the proportion of bound holes increased; this indicates that the constraints of the rock particle skeleton were completely broken at this stage, and a large number of secondary fine pores were generated. The decrease in the proportion of seepage holes was due to the fact that the test did not make up water, and the water stored in the seepage holes migrated to the bound pores, but this does not imply that the actual seepage holes decreased.

## 4. Fractal Characteristics of $T_2$ under Freeze–Thaw Cycles

### 4.1. $T_2$ Fractal Theory of NMR

The goal of fractal theory is to investigate the geometric distribution of pores using dimensional perspective and mathematical methods, which can reflect the effectiveness of space occupied by complex shapes and unify order and disorder in nonlinear systems. Because freezing–thawing damage to rock includes the evolution process of pore development, the change in pore space characteristics during the freeze–thaw cycles can be quantitatively analyzed using fractal theory.

If the pore size distribution of rocks has fractal characteristics, the approximate NMR geometric fractal formula can be used as shown in Equation (1) [35]:

$$\text{Lg}(W) = (3 - D)\,\text{Lg}(T_2) + (D - 3)\,\text{Lg}(T_{2\text{max}}) \tag{1}$$

where $W$ is the percentage of cumulative pore volume with transverse relaxation time less than $T_2$ in total pore volume; $D$ is the fractal dimension; and $T_2$ max is the transverse relaxation time corresponding to the maximum diameter.

Fitting the $T_2$ NMR spectrum yields Equation (2), and the slope $a$ of the fitting formula can obtain the fractal dimension; $b$ is the intercept.

$$\text{Lg}(W) = a\,\text{Lg}(T_2) + b \tag{2}$$

That is, the pore fractal dimension obtained based on the NMR $T_2$ spectrum is:

$$D = 3 - a \tag{3}$$

### 4.2. Analysis of Pore Fractal Characteristics

The fitting analysis of the fractal dimension of the full scale domain, according to Equation (2), showed that the fitting degree is between 0.543 and 0.591, which was too low, and the results could not accurately characterize the change in the pore structure of the full-scale domain. As a result, the full-scale pores were divided into bound pores and seepage pores based on the difference in pore characteristics, with the $T_2$ cut-off value of 5.264 ms obtained above as the boundary, and the fractal dimension was calculated. Simultaneously, the fractal characteristics were examined, and the fitting degree was 0.728~0.837. Figure 8 depicts the results using the initial state as an example, as well as the fractal dimension changes of the two types of pores at different cycles.

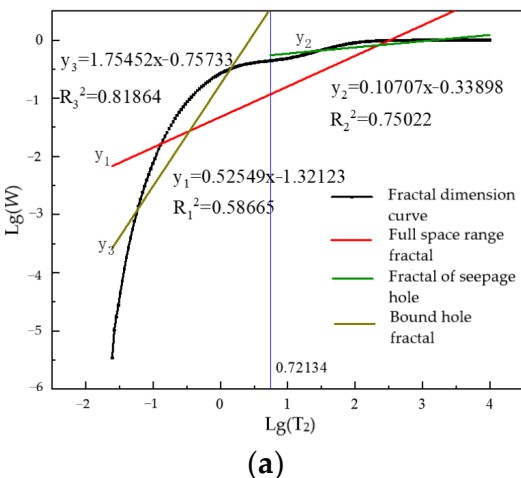

(**a**)

**Figure 8.** *Cont.*

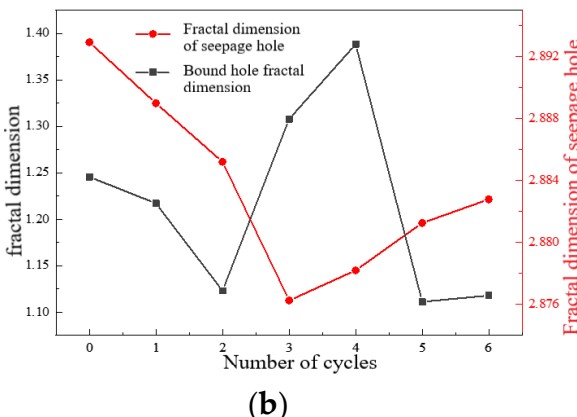

**(b)**

**Figure 8.** (**a**) Fractal dimension of initial state; (**b**) fractal dimension variation curves of various holes before and after freeze–thaw cycles.

The fractal theory states that a change in fractal dimension can indicate a change in the direction of rock pore structure. When the fractal dimension increases, the homogeneity of the pore structure worsens and the pore surface becomes rough; when the fractal dimension decreases, the homogeneity of the pore structure improves and the pore surface smoothens [21]. According to the results of the tests, it can be seen that:

(1) The fractal dimensions of the seepage hole and the bound hole decreased significantly in the first two cycles, indicating that the homogeneity of the seepage hole improved and the pore throat surface became more regular.

Although the rock produced some pore expansion at this stage, this expansion effect only occurred within the original pore size. There was no associated pore expansion, and the frost heaving effect failed to break through the constraints of the rock particle skeleton.

(2) The fractal dimension of the seepage hole continued to decrease sharply in the third cycle, while the fractal dimension of the bound hole increased significantly. The homogeneity of the seepage hole was greatly improved after the third freeze–thaw cycling. The constraint of the rock particle skeleton was reduced after the freeze–thaw cycling, and the frost heaving effect was intensified, resulting in the gradual expansion of the pores within the scale of the seepage hole; at the same time, the homogeneity of the bound pores became worse in this process. It can be seen that new pores on the bound pore scale were generated during the third freezing–thawing cycle.

The rock skeleton deteriorated after three freeze–thaw cycles, resulting in a surge of pressure in the bound hole after the water migrated to the wall of the seepage hole during freezing, thus breaking the constraint of the rock skeleton, transitioning from the bound hole to the seepage hole, and producing secondary bound holes.

(3) The fractal dimension of the seepage holes increased continuously by a small margin in subsequent cycles, whereas the fractal dimension of the bound holes increased only in the fourth cycle, but decreased significantly in the fifth cycle, and the fractal dimension of the bound holes changed very little in subsequent cycles. The new secondary pores in the bound pore scale increased significantly during the fourth freezing and thawing cycle, while the conversion rate to the seepage pore scale slowed. The pores within the bound pore scale were fully developed during the fifth freeze–thaw cycle, and a small amount of bound pores changed to the seepage pore scale. The bound pore and seepage pore did not change significantly during the sixth freezing–thawing cycle.

The above phenomena show that, as a result of damage accumulation caused by freezing and thawing, the binding effect of a rock particle skeleton on water migration during the freezing process was completely broken through after the fourth freezing and thawing cycle, resulting in a sharp increase in pores within the bound pore scale at this stage. In the fifth freezing–thawing cycle, the bound pore further transformed to the seepage pore size, but the growth rate slowed, and the new pore structure was similar

to the original seepage pore structure. It should be noted that the increase in the fractal dimension of the seepage hole after the third freezing–thawing cycle might be due to the migration of internal water to the bound hole, which does not imply that the homogeneity of the real seepage hole was reduced.

To summarize, the analysis of pore structure during the freezing–thawing cycle showed that the degradation of weakly cemented sandstone by freezing–thawing can be divided into three stages when no water supplement is present: during the first two freezing–thawing actions, water migrates to the pore wall within the seepage pore scale, resulting in pore expansion within this scale. There are few secondary pores due to the constraint of the rock skeleton. After two freezing-thawing cycles, the constraint of the rock skeleton is reduced. After water migrates to the pore wall within the seepage pore scale during the third and fourth freezing–thawing cycles, it continues to migrate to the pores within the bound pore scale, resulting in high internal pore pressure, which opens the constraint of the rock skeleton and generates secondary pores. Thereafter, because no additional water is added to the pores within the seepage pore scale, they become unsaturated, and most of the pores within the bound pore scale become saturated. The degree of homogenization increases. It is difficult to generate higher pore internal pressure during the freezing process, and it is also difficult to reinforce the constraints of the rock skeleton so that no new pores form.

## 5. Characteristics of $T_2$ Peak Splitting under Freeze–Thaw Cycles

### 5.1. NMR Pore Size Division and Processing Method

#### 5.1.1. NMR Pore Size Division

It is commonly assumed in NMR $T_2$ analysis that pores of the same size only show a fixed value in the $T_2$ spectrum, and that the $T_2$ spectrum can be obtained by accumulating this fixed value of pores of all sizes in the sample. However, according to the NMR testing principle, the decay rate of the hydrogen nucleus is related to the degree of binding on the fluid, and the further the water inside the pore is from the pore surface, the lower the binding degree, implying that the actual test result of a single pore should be in the wave crest area (Figure 9).

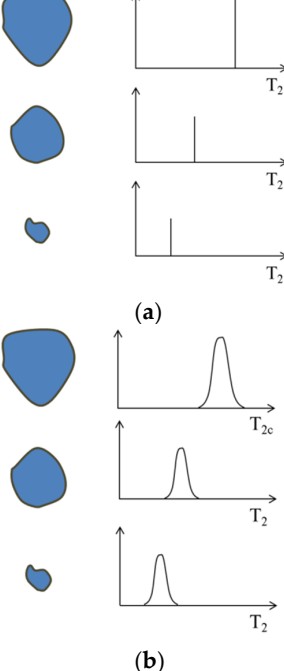

**Figure 9.** (**a**) Analysis mode of NMR $T_2$ under ideal conditions; (**b**) analysis mode of NMR $T_2$ in its actual state.

Furthermore, the deterioration in the pore structure of weakly cemented sandstone is represented by the transformation between different scales of pores, according to the above analysis. In light of this, the $T_2$ peak splitting method can be used to narrow the pore division range and then obtain the pore evolution characteristics of different scales, in order to further analyze the transformation rule between different scales of pores during freezing–thawing cycles and to reveal the evolution mechanism of meso freezing–thawing damage in more depth.

The $T_2$ spectrum measured in this experiment had a bimodal distribution, and the peak fitting results had multiple solutions. As a result, while ensuring a high degree of fitting, it is necessary to consider the practical significance of the fitting results. Therefore, it was necessary to choose the smallest number of peak splittings.

Multiple fitting analysis revealed that when the number of peak splittings is less than four, the fitting degree is less than 0.8. As a result, the number of fitting peaks was determined to be four.

The peak center point and half peak width were determined based on the analysis results of bound pores and seepage pores. The data are shown in Table 2.

**Table 2.** Selected fitting peak center and half-peak width.

| Peak | Peak Center (ms) | Half-Peak Width (ms) |
| :---: | :---: | :---: |
| 1 | 0.488 | 0.384 |
| 2 | 1.589 | 1.244 |
| 3 | 15.703 | 12.522 |
| 4 | 54.789 | 44.436 |

5.1.2. Processing Method of Peak Splitting

A single peak can be described by a Gaussian function. The independent variables of the $T_2$ spectrum were first converted using a log function, and then fitted using the least square method. Equation (4) depicts the Gaussian function.

$$y = y_0 + \frac{A e^{\frac{-4\ln(2)(x-x_c)^2}{w^2}}}{w\sqrt{\frac{\pi}{4\ln(2)}}} \tag{4}$$

where $y_0$ is the baseline, its value is 0 in $T_2$ curve; $x_c$ is the selected peak center; $A$ is the peak area; and $w$ is the selected half-peak width.

*5.2. Peak Splitting Analysis of Pore*

Figure 10 depicts the peak splitting curve and fitting curve of the $T_2$ spectrum in its initial state, and Table 3 depicts the peak height parameters of the $T_2$ spectrum in each experimental state, as well as the fitting degree of the corresponding fitting curve.

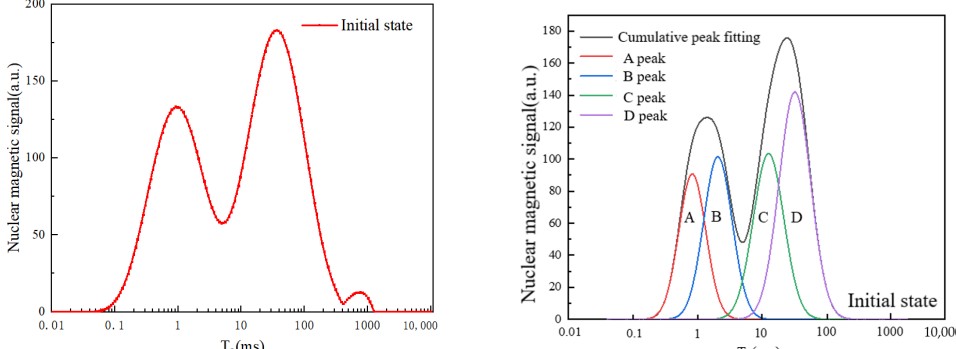

**Figure 10.** Initial $T_2$ spectrum and peak splitting results before freeze–thaw cycles.

**Table 3.** Fitting degree and peak height of each wave peak.

| | Number of Cycles | A Crest | B Crest | C Crest | D Crest | Degree of Fit (%) |
|---|---|---|---|---|---|---|
| Peak height | 0 | 90.784 | 101.479 | 103.548 | 141.959 | 99.765 |
| | 1 | 92.749 | 108.714 | 90.065 | 167.824 | 99.670 |
| | 2 | 88.429 | 105.401 | 84.361 | 168.092 | 99.685 |
| | 3 | 92.971 | 72.600 | 96.993 | 138.538 | 99.929 |
| | 4 | 93.125 | 78.784 | 93.756 | 147.266 | 99.902 |
| | 5 | 88.665 | 98.723 | 85.167 | 162.382 | 99.720 |
| | 6 | 89.577 | 98.239 | 85.081 | 161.522 | 99.703 |

As shown in Figure 11, the proportion of peak area and total spectral area of each peak were calculated under different freezing–thawing cycles.

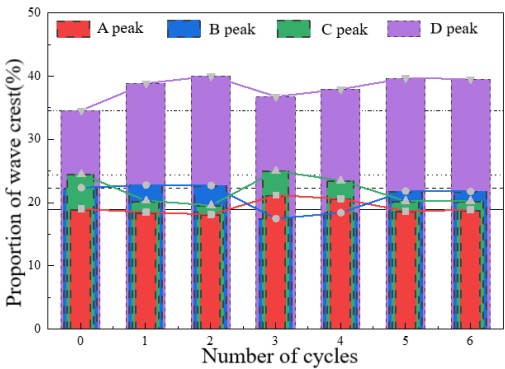

**Figure 11.** Proportion of each wave peak as the number of freeze–thaw cycles increases.

The figure shows that, due to the limitations of the rock's initial pore distribution, the proportion of the D peak was always the greatest, while the proportions of the other three peaks increased and decreased during the freezing–thawing cycles. The test results can be divided into three stages.

(1) The proportion of B and D peaks increased in the first two cycles, while the proportion of A and C peaks decreased. Assuming that the pore can only gradually change to a larger scale and that the transformation is irreversible, it can be deduced that the pore with peak A gradually expanded to the pore with peak B in this process, and the pore with peak C expanded to that with peak D. When combined with the fractal analysis results, it became clear that the reason was that the constraint force of the rock particle skeleton was large at this stage, and the first two freezing–thawing effects only led to the expansion of pore surface shapes in the bound pores and seepage pores.

(2) The proportion of peaks B and D decreased during the third freezing–thawing cycles, while the A and C peaks increased. The cumulative damage caused by freezing and thawing weakened the constraint effect of the rock particle skeleton around the bound pore, causing the pore where peak B was located to expand to peak C at this stage. More secondary pores were generated, leading to an increase in the proportion of peak A. Because there was no water supplement in the pores within this range, the proportion of peak D decreased.

(3) The proportions of peaks B and D gradually increased and remained unchanged in the following cycles, while the proportions of peaks A and C gradually decreased and remained unchanged. According to the pore transformation assumption, the pore where peak A was located gradually expanded towards the pore where peak B was located in the fifth freezing–thawing cycles, and the degree to which peak B expanded into peak C was less than the new increment of peak B. At the same time, the degree to which peak C expanded into peak D was greater than that to which peak B expanded into peak C. At the same time, the degree to which peak C expanded into peak D was greater than that of peak B expanding into peak C.

Based on the above pore structure research, it can be seen that after three freezing-thawing cycles, the cumulative damage of weakly cemented sandstone reduced the confinement effect of the particle skeleton around the pores, and the frost heaving effect caused by water migration was able to easily damage the pores. Pore expansion occurred in all types of pores; and when pore connectivity increased to a certain extent, the role of water migration and frost heaving could not be enhanced any further without water supplement. The freeze–thaw cycling no longer caused visible pore expansion, and the proportion of pores tended to be stable on all scales.

To summarize, we discovered that the freezing–thawing degradation of weakly cemented sandstone has a stage characteristic, and the strength of its freeze–thaw cycle degradation effect is dependent on the weakening of the rock skeleton strength under freeze–thaw cycles as well as on the connectivity of small-scale pores in the rock. The lower the rock skeleton strength, the worse the pore connectivity and the more obvious the freeze–thaw degradation effect, and vice versa.

## 6. Conclusions

(1) The evolution of the pore structure of weakly cemented sandstone can be divided into three stages under the action of freezing and thawing cycles. Stage 1: the rock skeleton can still significantly constrain frost heave and the pore expansion effect of rock only occurs within the primary pore scale, mainly showing the transformation between adjacent scales. Stage 2: as the freezing and thawing cycles increase, the constraint of the rock skeleton on frost heave decreases, small-scale secondary pores emerge gradually, pore expansion occurs gradually, and pore connectivity increases gradually. Stage 3: with the increase in the connectivity of the rock pores, the effect of the pores' internal pressure growth in the freezing process caused by water migration is weakened. Thus, the skeleton constraint cannot be broken, and freezing and thawing can no longer have an obvious expansion effect on the rock's pore structure.

(2) The freezing–thawing deterioration of rock has a stage characteristic, and the strength of its freeze–thaw cycling deterioration effect could be determined by the weakening of the rock skeleton's strength during the freezing–thawing cycles, as well as by the connectivity of small pores in the rock. The lower the rock skeleton strength, the worse the pore connectivity and the more visible the freezing–thawing deterioration effect, and vice versa.

(3) Without water supplement, weakly cemented sandstone undergoes three freezing–thawing deterioration before the bound hole can be transformed into a seepage hole. The condition of gradual pore expansion requires the rock to undergo more than four freezing–thawing deteriorations, and the freeze–thaw cycling did not produce an obvious pore expansion effect after five cycles of deterioration.

**Author Contributions:** Conceptualization, J.L. and Y.Y.; methodology, J.Y. and Y.L.; software, Y.Y. and X.L.; formal analysis, J.L., J.Y. and Y.Y.; writing—original draft preparation, Y.Y. and J.Y.; writing—review and editing, J.L., Y.Y. and J.Y. All authors have read and agreed to the published version of the manuscript.

**Funding:** The financial support by the Natural Science Foundation of Anhui (Grant No.2208085QE142, No.2008085ME165); The Opening Foundation of Anhui Province Key Laboratory of Building Structure and Underground Engineering (KLBSUE202103) and The Opening Foundation of Ministry of Education Engineering Research Center of Underground Mine Construction (JYBGCZX2020102).

**Data Availability Statement:** The data presented in this study are available upon request from the corresponding author.

**Acknowledgments:** The authors are grateful to executive editors and anonymous journal reviewers for their helpful comments and suggestions on revisions of this paper, and to Cheng hua for his time advising on the manuscript.

**Conflicts of Interest:** The authors declare no conflict of interest.

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
