# Peer review of "Study on Pore Structure Evolution Characteristics of Weakly Cemented Sandstone under Freeze–Thaw Based on NMR"

_water, doi:10.3390/w15020281_

Round 1
Reviewer 1 Report
The authors may consider the following improvements:
1. type cm3 correctly with the superscript.
2. Use acronym if it is properly defined, e.g., Line 71 NMR
3. Fig. 1, the horizontal line presents the average at 736.9 m/s, but Table 1 presents the the wave velocity to be 733 m/s. Should these numbers be consistent?
Reviewer 2 Report
Comments on Water- 2063817
The manuscript is unintelligible and may be misleading because the authors lack logical thinking. Any challenging report is acceptable if it is well written and provides necessary references. A pile of undigested materials and muddled symbols only degrade the readability. I list a few problems for the authors to consider.
1. The abbreviations adopted in the manuscript and the way of their appearances are not following common practice. Check the journal’s guidelines or any good writers’ handbook to make corrections. Also notice the different conventions of using abbreviations in the abstract and main text; the authors should pay attention to this issue.
2. Provide some details regarding the field procedure of taking samples. Usually the environments and parameters i.e. confining pressure, temperature, moisture, etc. alter drastically during a drilling operation. How do the authors believe that they were able to maintain the in-situ environments?
3. Many statements in the main text are ambiguous and illogical. For instance, the nomenclatures “original pore types” appear only once, and the authors explained that they are divided by 0~3ms, 3~33ms, 33~505ms. Those data sound like three different periods in milliseconds, but the authors declared there are four types of pores. Should the authors reorganize their thoughts and make a more intelligible statement?
4. Equations 1 to 4 are meaningless; mainly the symbols in the equations are weird, what do Eq 21(1) and 1g indicate? What do the constants (parameters) a and b come from? In addition, Equation 4 needs precise citation or derivations if the authors modified it.
Round 2
Reviewer 2 Report
Comments on Water- 2063817 V2
Although some typos, basic grammatical issues, and format inconsistency in references remain, the manuscript is acceptable to publish in Water, as the English editing of MDPI is supposed to deal with those problems.
Reviewer 3 Report
The authors have addressed the questions generally and made changes accordingly.